# Understanding Primary Care Physician Vaccination Behaviour: A Systematic Review

**DOI:** 10.3390/ijerph192113872

**Published:** 2022-10-25

**Authors:** Ángela Prieto-Campo, Rosa María García-Álvarez, Ana López-Durán, Fátima Roque, Maria Teresa Herdeiro, Adolfo Figueiras, Maruxa Zapata-Cachafeiro

**Affiliations:** 1Department of Preventive Medicine and Public Health, University of Santiago de Compostela, 15786 Santiago de Compostela, Spain; 2Department of Preventive Medicine and Public Health, Hospital of Santiago de Compostela, 15706 Santiago de Compostela, Spain; 3Department of Clinical Psychology and Psychobiology, University of Santiago de Compostela, 15786 Santiago de Compostela, Spain; 4Research Unit for Inland Development, Polytechnic of Guarda (UDI-IPG), Avenida Dr. Francisco Sá Carneiro, No. 50, 6300-559 Guarda, Portugal; 5Health Sciences Research Centre, University of Beira Interior (CICS-UBI), Av. Infante D. Henrique, 6200-506 Covilhã, Portugal; 6Escola Superior de Saúde, Instituto Politécnico da Guarda Rua da Cadeia, 6300-035 Guarda, Portugal; 7Institute of Biomedicine (iBiMED), Department of Medical Sciences, University of Aveiro, 3810-193 Aveiro, Portugal; 8Consortium for Biomedical Research in Epidemiology and Public Health (CIBER of Epidemiology and Public Health, CIBERESP), Instituto de Salud Carlos III, 28029 Madrid, Spain; 9Health Research Institute of Santiago de Compostela (IDIS), 15786 Santiago de Compostela, Spain

**Keywords:** vaccination hesitancy, primary care physician, knowledge, perception, attitude, public health

## Abstract

Background: Vaccine hesitancy decreases adult vaccination coverage and has been recognized by WHO as a major health threat. Primary care physicians (PCP) play a key role in vaccination by giving vaccine counselling to their patients. The aim of this systematic review is to identify the knowledge, beliefs, attitudes and barriers (KBAB) associated with own vaccination and patient recommendation in primary care physicians. Methods: MEDLINE/PubMed, EMBASE and Cochrane Library databases were used to search and identify relevant studies based on their title and abstract. In the next step, the full text of each previously selected article was read for eligibility. Articles were selected by two independent reviewers and data extraction was performed using tables. The following information was extracted: methodological characteristics, demographic factors, professional characteristics, and intrinsic or extrinsic factors influencing vaccination or recommendation. Results: Our search yielded 41 eligible papers, data-sources, previous practices, belief in the effectiveness or safety of the vaccine, perceived risk, and trust in health authorities were all shown to be related to own vaccination and patient recommendation. Conclusion: Internet is the main source of information for PCP related to vaccine hesitancy. It is therefore essential to increase the presence and access to pro-vaccination content in this area. In addition, involving PCP in the establishment of vaccination recommendations could improve their credibility in the institutions. On the other hand, training in communication skills and establishing reminder systems could reflect higher vaccination coverage among their patients.

## 1. Introduction

Vaccines rank among the greatest advances in world health, indisputably preventing over two million deaths per year [1]. In fact, it is the tool that reduces the most deaths from disease, second only to the introduction of safe drinking water [2]. Historically, immunization programs have been targeted at children, a strategy that has achieved notable success in the control of infectious diseases (smallpox, polio, etc.) [3], due to the extensive vaccine coverage achieved.

That said, however, in the case of vaccines targeted at the adult population, coverage has been far lower [4,5]. A large part of this low coverage among adults is attributable to vaccine hesitancy. This is a situation of doubt, which could be resolved toward pro-vaccination or anti-vaccination (dragged by the anti-vaccination movements) [6]. At present, the magnitude of the problem is such that in 2019 the World Health Organization (WHO) included it as one of the main threats to global health [7]. In addition, combating vaccine hesitancy is also a challenge for the WHO’s Immunization 2030 Agenda [8].

Primary care physicians (PCP) play a key role in the vaccination of adults [9,10,11]: on the one hand, their own immunization is important for their personal protection and that of their patients; and on the other hand, motivated physicians have been seen to be more effective when it comes to vaccinating their patients [12,13]. The role of the PCP in addressing vaccine hesitancy is decisive, as they are the first and most reliable source of information for patients when deciding whether to be vaccinated [14]. The aim of the study is to identify the factors (knowledge, beliefs, attitudes and barriers) that condition the vaccination of PCP and also the vaccination recommendations to their adult patients [15,16,17,18].

## 2. Material and Methods

### 2.1. Study Protocol and Registration

The review was conducted following the Preferred Reporting Items for Systematic Reviews and Meta-Analyses (PRISMA) guidelines [19]. The protocol was registered in the PROSPERO International Register of Systematic Reviews (registration no. CRD42021227730).

### 2.2. Search Strategy 

For the purpose of this systematic bibliographic review, we conducted a search in MEDLINE/PubMed, EMBASE and the Cochrane Library electronic databases covering the period 1 January 2011 to 6 November 2021. The search terms used to identify relevant papers is presented at Appendix A.

### 2.3. Inclusion and Exclusion Criteria

Studies were considered eligible for review if they met the following criteria: (i) they had been published in English, Spanish or Portuguese; (ii) they were quantitative or mixed (considering only quantitative data); (iii) they sought to explore and identify the KBAB of PCP in relation to any adult vaccine; (iv) they assessed any association between KBAB and own vaccination and patient recommendation; and (v) their study population included any physicians performing primary care functions (family physicians/general practitioners/general internists/obstetrician gynecologists) but excluded medical residents. Furthermore, physicians working in a hospital setting were required to have spent at least 50% of their time in primary care.

Following deduplication, titles and abstracts were screened by two authors (RG, AP), working independently. All papers identified as potentially relevant were reviewed by the authors, and in the event of disagreement, the paper in question was examined by AF and MZ, who then took the final decision.

### 2.4. Data-Extraction and Analysis

For each study included in the review, a table was drawn up (see Table 1), with the following parameters: author; year; country; study population; number and type of primary care participants; response rate (%); vaccine; data-collection method; and final Newcastle-Ottawa Scale (NOS) score [20]. We have covered all vaccines used in adults, but the adult vaccination recommendations vary from country to country. However, influenza and HPV vaccines are recommended in many countries, so we have structured our tables and results according to: influenza, vaccine, HPV vaccine or adult vaccines in general. A second table also was likewise created (Table 2) showing the following socio-demographic factors and their influence on own vaccination and patient recommendation: age; gender; employment status; experience; number of patients; type of practice; and practice of alternative medicine.

In addition, the intrinsic and extrinsic factors reported by each study and their influence on own vaccination and patient recommendation (if any) were extracted and, respectively, listed in a third table (Table 3).

For studies that performed statistical hypothesis testing between the KBAB model and own vaccination and patient recommendation, we collected the relevant odds ratios (OR), and in any case where these were not available, the *p*-values and percentages. For results purposes, vaccines were classified into three groups: vaccines in general (studies that included a number of vaccines); influenza vaccine; and human papillomavirus (HPV) vaccine. This classification was chosen because a large proportion of the published studies included in our review addressed only influenza vaccine, a large proportion included only HPV vaccine, and the remaining studies addressed several vaccines at the same time. Data were extracted by two authors (RG, AP). Differences of opinion were resolved by discussion between the two authors, and if no agreement could be reached, it was left to AF and MZ to decide the matter.

### 2.5. Quality Assessment

To assess risk of bias in the studies selected for inclusion, we used the Newcastle-Ottawa Scale adapted for cross-sectional studies [20]. Two authors (RG, AP) independently assessed the quality of the studies. Any differences of opinion were resolved by consensus, and in any case where consensus could not be reached, the paper was then examined by AF and MZ.

**Table 1 ijerph-19-13872-t001:** Methodological characteristics of the papers selected.

Author	Year	Country	Study Population	Number and Type of Primary Care Participants	Response Rate (%)	Vaccine	Data-Collection Method (Questionnaire)	Final NOS Score
**With statistical hypothesis testing between** **KBAB and own vaccination and patient recommendation**								
Verger P et al. [21]	2021	France	GP	2755 GP	43.9 T	COVID-19	Online	7
Verger P et al. [22]	2021	France	GP	2755 GP	29 T	COVID-19	Online	7
Arlt J et al. [23]	2021	Germany	GP	308 FM, 111 IM, 24 WS (443)	28.0 E	Influenza	Mail	7
Neufeind J et al. [24]	2020	Germany	FP	735 FP	20.4 E	Adult vaccination: influenza, pertussis, hepatitis B, measles, DT	Phone	6
Verhees RAF et al. [25]	2020	Netherlands	GP	552 GP	31.7 E	Influenza	Online	5
Vezzosi L et al. [26]	2019	Italy	GP	73 GP	26.6 T	Adults ≥ 65: Seasonal Influenza (SI), Pneumococcus (PNV), Zoster (ZV)	Online	3
Yilmaz-Karadağ F et al. [27]	2019	Turkey	PCP	49 FP, 172 GP (221)	14.7 T	Adult risk groups: Influenza, Td, pneumococcal, meningococcal, HBV, HAV	Online	5
Akan H et al. [28]	2016	Turkey	FP	606	27.5 E	Influenza	Online	4
Klett-Tammen CJ et al. [29]	2016	Germany	GP, PA	774 GP	13.4 E	Vaccines for the elderly: tetanus, influenza, pneumococcal	Mail	6
Verger P et al. [30]	2015	France	GP	1582 GP	42.5 E	Influenza, hepatitis B, MMR, HPV, MenC,	Telephone	6
Flicoteaux R et al. [31]	2014	France	GP	1431 GP	36.8 E	Pandemic Influenza A/H1N1	Telephone	4
Pulcini C et al. [32]	2013	France	GP	1431 GP	36.8 E	Hepatitis B, pertussis, seasonal and pandemic influenza	Telephone	5
Pulcini C et al. [33]	2014	France	GP	329 GP	36.3 E	MMR	Telephone	3
Verger P et al. [34]	2012	France	GP	1431 GP	36.8 E	Pandemic Influenza A:H1N1	Telephone	6
**NO** **statistical hypothesis testing between** **KBAB and own vaccination and patient recommendation**								
Deruelle et al. [35]	2021	USA	GP, OB/GYN	69 GP	81.1 T	COVID-19	Online	5
Bayliss J et al. [36]	2021	Australia	GP, AC	412 GP	-	Adult vaccination (focus: pertussis) Influenza, Td, HBV, HBA, polio	-	3
Hurley LP et al. [37]	2021	USA	PCP	336 FP, 281 GIM (617)	64.0 E	HPV	Mail or online	4
Napolitano F et al. [38]	2021	Italy	GP	349 GP	61.5 E	HPV	Online or Telephone	7
Celep G et al. [39]	2020	Turkey	PHW	97 FP	-	Pregnancy (Td, Tdap, HBV, influenza)	Online or Telephone	3
Kalemaki D et al. [40]	2020	Greece	GP	260 GP	88.0 E	Influenza, measles, HBV, pertussis, Tdap	Online or Telephone	4
Meites E et al. [41]	2020	USA	PCP	266 FM, 235 GIM (430)	59.3 E	HPV	Mail or online	6
Awadlla NJ et al. [42]	2019	Saudi Arabia	PHW	74 PCP	77.0 E	Seasonal Influenza	Personally	6
Collange F et al. [43]	2019	France	GP	2586 GP	37.6 E	MMR, MenC, HBV, seasonal influenza, HPV	Telephone	5
Glavier M et al. [44]	2019	France	GP	287 GP	21.9 E	Vaccination practices with chemotherapy patients: influenza, pneumococcal, DTP	Fax, mail or online	5
Hurley LP et al. [45]	2018	USA	PCP	FP, GIM (603)	65.0 E	Zoster Vaccine Live (ZVL) and New Recombinant Zoster Vaccine (RZV)	Mail or online	5
Le Marechal M et al. [46]	2018	France	GP	1582 GP	42.5 E	Seasonal Influenza, HBV, HPV	Telephone	5
Levi M et al. [47]	2018	Italy	GP	1245 GP	12.4 E	Seasonal Influenza	Online	4
Merriel SWD et al. [48]	2018	UK	GP, SHCP	38 GP	-	HPV	Online	5
Steben M et al. [49]	2018	Canada	GP, OB/GYN	378 GP	8.0 E	HPV	Online	5
Desiante F et al. [50]	2017	Italy	GP	229 GP	48.6 E	Influenza	Online	3
Hurley LP et al. [51]	2017	USA	PCP	317 GIM, 236 FP (553)	66.4 E	Adult vaccination: zoster, hepatitis B, Tdap, hepatitis A, HPV, Meningococcal, chicken pox, Td, PVC13, seasonal influenza, PPSV23, MMR	Mail or online	4
Hurley LP et al. [52]	2016	USA	PCP	352 GIM, 255 FP (607)	71.8 E	Adult vaccination: seasonal influenza, pneumococcal, Tdap, Td, herpes zoster, MMR	Mail or online	7
Raude J et al. [53]	2016	France	GP	1582 GP	92.4 E	MMR, MenC, HPV, HBV, seasonal influenza	Telephone	7
Verger P et al. [54]	2016	France	GP	1582 GP	46 E	Influenza, dTP, HBV, MMR, MenC, HPV	Telephone	6
Massin S et al. [55]	2015	France	GP	1136 GP	29.2 E	Influenza	Telephone	3
Alsaleem MA [56]	2013	Saudi Arabia	PHW	95 PCP	81.2 E	H1N1 vaccine	-	4
François M et al. [57]	2011	France	FP	341 FP	17.0 E	Hepatitis B	Online	4
Inoue Y et al. [58]	2011	Japan	GP	515 GP	51.5 T	Novel Pandemic Vaccine Influenza A/H1N1	Mail	6
Lutringer-Magnin D et al. [59]	2011	France	GP	279 GP	93.0 T	HPV	Mail	6
Rurik I et al. [60]	2011	Hungary	FP	198 FP	85.0 E	Pandemic Influenza	Personally	3
Ward K et al. [61]	2011	Australia	HCW	79 GP	36.0 E	Annual Influenza Vaccination	Mail	4

NOS: Newcastle-Ottawa Scale. Study population: AC: Adult consumer; GP: General Practitioner; HCW: Healthcare worker; PCP: Primary care physician; PHW: Primary healthcare worker; SHCP: Sexual healthcare professional. Number and type of primary care participants: FP: Family Physician; GIM: General Internal Medicine; WS: Without specialization. Response rate: E: Calculated from eligible subjects; T: Calculated from total subjects.

**Table 2 ijerph-19-13872-t002:** Demographic factors and professional characteristics.

Author	Age	Gender	Status	Experience (Years of Practice)	No. Patients	Type of Practice (Single/Group)	Occasional Practice Alternative Medicine
**Statistical hypothesis testing between** **KBAB and own vaccination and patient recommendation**							
Verger P et al. [21]		♀: ↓					
Verger P et al. [22]	>age: ↑	♀: ↓					
Arlt J et al. [23]							
Neufeind J et al. [24]	>age: ↓ [OV] >age: ↓ MEAS [PR]	♂: ↑ MEAS [PR]					Yes: ↓ INF [PR]
Verhees RAF et al. [25]	≥60: ↑ [OV]	♂: ↑ [OV]					
Vezzosi L et al. [26]							
Yilmaz-Karadağ F et al. [27]	31–40: ↑ INF + Td + HBV [OV] 31–40: ↑ INF + Td + PNV [PR]	♀: ↑ INF [OV] ♂: ↑ PNV + Td [PR]		≅	≅		
Akan H et al. [28]	>age: ↑ [OV]	≅		>years: ↑ [OV]			
Klett-Tammen CJ et al. [29]							
Verger P et al. [30]							
Flicoteaux R et al. [31]							
Pulcini C et al. [32]	<age: ↑ [OV]				>no.: ↑ [OV]	≅	Yes: ↓ [OV]
Pulcini C et al. [33]	≅	≅			≅	≅	≅
Verger P et al. [34]	≅	≅			≅	Group: ↑ [OV]	Yes: ↓ [OV]
**NO** **statistical hypothesis testing between** **KBAB and own vaccination and patient recommendation**							
Deruelle et al. [35]							
Bayliss J et al. [36]							
Hurley LP et al. [37]							
Napolitano F et al. [38]							
Celep G et al. [39]							
Kalemaki D et al. [40]	<age: ↑ MEAS [OV] <age: ↓ INF [OV]	♀: ↑ HBV [OV]					
Meites E et al. [41]							
Awadlla NJ et al. [42]							
Collange F et al. [43]							
Glavier M et al. [44]				≅			
Hurley LP et al. [45]			FP: ↑ ZVL [PR]			Small: ↑ ZVL [PR]	
Le Marechal M et al. [46]							
Levi M et al. [47]	>age: ↑ seasonal INF >age: ↓ Td [OV]	♂: ↑ seasonal INF, pandemic INF, PNV [OV]		>years: ↑ PNV >40 years: ↑ pandemic INF			
Merriel SWD et al. [48]							
Steben M et al. [49]		≅					
Desiante F et al. [50]	≅	♂: ↑ [OV]	≅		>no.: ↑ [OV]		
Hurley LP et al. [51]			≅				
Hurley LP et al. [52]							
Raude J et al. [53]							
Verger P et al. [54]	≅	≅				≅	Yes: ↓
Massin S et al. [55]	>age: ↑ pandemic INF [OV]	♀: ↓ seasonal INF [OV]			High: ↑ seasonal + pandemic INF [OV] pandemic INF [PR]	Group: ↑ seasonal + pandemic INF [OV]	Yes: ↓ seasonal + pandemic INF [OV]
Alsaleem MA [56]							
François M et al. [57]	>age: ↑ [PR]	≅			>3/day: ↑ [PR]		Yes: ↓ [PR]
Inoue Y et al. [58]							
Lutringer-Magnin D et al. [59]							
Rurik I et al. [60]							
Ward K et al. [61]							

INF (influenza); HPV (human papillomavirus vaccine); HBV (hepatitis B); MEAS (measles); MMR (measles, mumps and rubella); PER (pertussis); PNV (pneumococcal vaccine); PPSV23 (pneumococcal polysaccharide vaccine); ZV (zoster vaccine); RVZ (recombinant zoster vaccine); ZVL (zoster live-attenuated vaccine); TT (tetanus vaccine); Td (tetanus-diphtheria); Tdap (tetanus, diphtheria, pertussis). [OV]: Own vaccination. [PR]: Patient recommendation. ≅ factor was not statistically significant; **↑** Factor leads to a statistically significant increase in own vaccination, patient recommendation; ↓ Factor leads to a statistically significant decrease in own vaccination, patient recommendation; ♀: Women; ♂: Man.

**Table 3 ijerph-19-13872-t003:** Intrinsic and extrinsic factors reported by each study and their influence on OV&PR.

Author	Data and Data-Sources	Knowledge	Safety	Efficacy/Benefits	Perceived Risk	Trust	Protection	Important
Verger P et al. [21]	-	-	Safe OR: 1.93	-	No perceived risk OR: 0.47	Trust in institutions OR: 0.86	-	-
Verger P et al. [22]	-	-	Safe OR:0.27	-	Perceived risk OR: 7.56	Trust in institutions OR: 0.10	-	-
Arlt J et al. [23]	Co-workers OR: 2.26 Media News OR: 0.16	-	-	-	-	-	-	-
Neufeind J et al. [24]	Official Sources OR: 6.95 [INF] [PR]	-	Safe OR: 1.64 [INF] [PR] Safe OR: 1.28 [MEAS] [PR] Safe OR: 1.42 [INF, PER, HBV] [OV]	-	-	Trust in institutions OR: 1.43 [INF] [PR]	-	No importance OR: 0.71 [INF, PER, HBV] [OV]
Verhees RAF et al. [25]	-	-	-	Efficacy *p* = 0.00 [OV]	-	-	-	-
Vezzosi L et al. [26]	-	GAP OR: 0.07 [ZV] [PR]	-	-	-	-	-	-
Yilmaz-Karadağ F et al. [27]	-	-	-	-	-	-	-	-
Akan H et al. [28]	-	-	Safe 1.55 < OR < 2.45 Median (OR): 2.08	Benefits 5.18 < OR < 9.29 Median (OR): 8.73	Perceived Risk 1.77 < OR < 7.49 Median (OR): 4.27 [OV]	Trust in institutions OR: 2.52	Natural Protection OR: 0.39	-
Klett-Tammen CJ et al. [29]	-	-	-	-	-	-	-	-
Verger P et al. [30]	-	-	-	-	-	-	-	-
Flicoteaux R et al. [31]	Official Sources OR: 2.03 Media News OR: 0.57	-	Safe OR:0.17 [PR]	Efficacy OR: 0.28 [PR]	No Perceived Risk OR:0.6 [PR]	-	-	-
Pulcini C et al. [32]	Internet OR: 0.92 [INF]	-	-	-	-	-	-	-
Pulcini C et al. [33]	Official Sources OR: 7.37	-	-	-	No Dangerous disease OR: 0.32 [MEAS] [PR]	-	-	-
Verger P et al. [34]	Official Sources *p* < 0.0001 No Media News *p* < 0.0001	-	Safe *p* < 0.0001	Efficacy *p* < 0.0001	Perceived Risk, Dangerous Disease *p* < 0.0001	Trust in institutions *p* < 0.0001 [OV]	-	-
**Author**	**Responsibility**	**Attitude**		**Extrinsic Factors**		**Practices (Behaviors)**		**Experiences**
Verger P et al. [21]	-	-		-		-		-
Verger P et al. [22]	-	-		-		-		-
Arlt J et al. [23]	-	-		Organizational Factor OR: 4.31 [P] Patient Facilitator OR: 3.20		-		-
Neufeind J et al. [24]	Responsibility OR: 1.82 [INF, PER, HBV] [OV]	-		-		Vaccination History OR: 44.09 [INF] [R]		-
Verhees RAF et al. [25]	-	-		-		-		-
Vezzosi L et al. [26]	-	Attitude + OR: 13.67 [PNV, ZV] [R]		-		Vaccination History OR: 5.44 [PNV] [R] Vaccination History OR: 19.36 [ZV] [R]		Patient Experience OR: 6.61 [ZV] [R]
Yilmaz-Karadağ F et al. [27]	-	-		-		Vaccination History *p* > 0.05 [R]		-
Akan H et al. [28]	-	Attitude + 3.06 < OR < 10.93 Median (OR): 7.245		Organizational Factor 2.64 < OR < 13.75 Median (OR): 6.57		Vaccination History OR: 15.1 Reminder System OR:1.66		-
Klett-Tammen CJ et al. [29]	-	-		-		-		-
Verger P et al. [30]	-	-		-		Vaccination History OR: 2.95 [INF] [R] Vaccination History OR: 1.90 [HBV] [R]		-
Flicoteaux R et al. [31]	-	-		-		-		Patient Experience OR: 2.81 [R] PE (30.2%)
Pulcini C et al. [32]	-	-		-		Vaccination History 1.08 < OR < 3.08 Mediana (OR): 1.2 [P]		-
Pulcini C et al. [33]	-	-		-		Check vaccination status OR: 3.38		-
Verger P et al. [34]	-	-		-		-		-

[OV]: Own vaccination; [PR]: patient recommendation; INF (influenza); HPV (human papillomavirus vaccine); HBV (hepatitis B); MEAS (measles); MMR (measles, mumps and rubella); PER (pertussis); PNV (pneumococcal vaccine); PPSV23 (pneumococcal polysaccharide vaccine); ZV (zoster vaccine); RVZ (recombinant zoster vaccine); ZVL (zoster live-attenuated vaccine); TT (tetanus vaccine); Td (tetanus-diphtheria); Tdap (tetanus, diphtheria, pertussis).

## 3. Results

### 3.1. Search Results

The chosen strategy retrieved a total of 815 articles in MEDLINE/PubMed, 107 in Cochrane Library and 3 in EMBASE. After de-duplication, 802 studies were included. Examination of the abstracts led to 108 papers being selected for a reading of the full text. A total of 41 papers were finally included (Figure 1).

Screening process to identify the articles included in our systematic review.

### 3.2. Quality Assessment

Subject to the main limitations specific to cross-sectional studies, the studies generally displayed a similar quality. The median scores were as follows: scale item selection, 3/5; comparability, 0/2; and outcome, 2/3 (Table 1).

### 3.3. Characteristics of Selected Studies

The general characteristics of the studies are summarized in Table 1. The studies were drawn from four different continents, albeit mainly from Europe (*n* = 29): seven studies had been conducted in North America, three in Asia, and two in Oceania (Australia). The median sample size was 552, and the median response rate was 37.6% in papers that considered the eligible population.

All the studies selected were cross-sectional in nature, with 40 of them being quantitative and 1 classified as mixed [59]. Of the 38 papers, statistical hypothesis testing between KBAB and own vaccination and patient recommendation was found in 14 but not in the remaining 27.

The study population comprised general practitioners (*n* = 30), family physicians (*n* = 10), general internists (*n* = 5), and primary care physicians without specification (*n* = 2).

In terms of data-collection, all the studies used questionnaire. While 21 studies employed online questionnaires and 11 studies used mail questionnaires, 14 studies applied their questionnaire interviewing participants by telephone, 2 in person and 1 by fax.

### 3.4. Vaccines in General

Although we located 19 papers that examined KBAB in respect of multiple general vaccines, only 7 reported statistical hypothesis testing between the KBAB model and own vaccination and patient recommendation.

#### 3.4.1. Studies without Statistical Hypothesis Testing between KBAB and Own Vaccination and Patient Recommendation

We found that the main data sources consulted were official and scientific information resources (*n* = 5), such as vaccination schedules. In some cases, PCP reported that the information received was inadequate. As regards the level of knowledge, as many as 8 papers reported shortfalls, due to ignorance of the composition of the vaccine or even of its very existence, misconceptions or lack of knowledge about health insurance cover in some countries, such as the USA.

With reference to vaccine safety, of all the papers which assessed this perception (*n* = 11), it was only in five that more than half of the PCP considered vaccines safe, since, in general, health professionals feared adverse reactions or long-term complications.

Similarly, when the perceived efficacy/benefits of vaccination were assessed in 11 papers, the participants were found to harbor many doubts about the utility of vaccines. Only in three studies did over half of the PCP believe in the benefits of the vaccine and its capacity to reduce complications. With respect to perceived risk (*n* = 10), 13% to 29% of physicians felt that there was no danger of suffering a vaccination-preventable disease and believed that they were not susceptible, except in 5 studies in which fear of suffering some chronic disease or presenting with some risk factor predominated. The importance attached by participants to vaccination (*n* = 5) was likewise found to be low: PCP showed themselves to be skeptical about the need for vaccination and did not consider it a priority.

Physicians’ attitudes toward vaccines were described in seven papers, which generally reported positive and favorable attitudes.

Certain external factors were seen to act as a barrier in the own vaccination and patient recommendation decision, with seven studies pinpointing organizational or logistical difficulties linked to the healthcare system, five papers detecting conflicts when it came to convincing the patient, and two studies identifying cost-related problems.

PCP’s previous practices were recorded in 10 papers. In most cases, these consisted of discussing vaccination with the patient, ascertaining patients’ vaccination status, using vaccine reminder systems, and storing vaccines.

#### 3.4.2. Studies with Statistical Hypothesis Testing between KBAB and Own Vaccination and Patient Recommendation

The studies that cited data sources (*n* = 4) showed that browsing the Internet was associated with a lower recommendation of vaccination, in contrast to what occurred when official or scientific sources were accessed.

PCP’s previous own and routine vaccine practices had an impact on their recommendations (*n* = 5). Thus, their own proactive behavior vis-à-vis vaccination, the fact of being active in the recommendation of vaccines in general and making a practice of ascertaining their patients’ immunization status, all had a positive influence on own vaccination and patient recommendation.

The occasional practice of alternative medicine was associated with lower acceptance of vaccines in two papers.

### 3.5. Influenza Vaccines 

A total of 12 papers examined KBAB associated with the influenza vaccine, with statistical hypothesis testing for the KBAB model and own vaccination and patient recommendation being reported in five.

#### 3.5.1. Studies without Statistical Hypothesis Testing between KBAB and Own Vaccination and Patient Recommendation

The data sources consulted were diverse and were described in four papers. In two of these, the participants reported receiving very little information from the relevant public institutions. The level of knowledge among PCP was not adequate: proof of this is that gaps or misconceptions were identified in five papers.

While vaccine safety was questioned in eight papers, 40% to 90% of PCP considered the vaccine safe in only two, and PCP reported doubts about the efficacy and benefits of vaccines in another three. Whereas the perceived risk (*n* = 8) of being susceptible or falling ill was seen as non-existent by most participants in four papers and in another four, one or more PCP was found to be risk averse. We located five papers which referred to the protection offered by the vaccine. The participants attached greatest value to personal protection, followed by protection of their patients, relatives, and friends. The trust placed by PCP in pharmaceutical companies or in reported data was found to be scant, which can be explained by previous studies that have shown that healthcare professionals believe that the pharmaceutical industry seeks financial gain through the sale of its products and control in clinical trials, rather than patient safety [62,63].

External factors that might act as drawbacks for acceptance of the vaccine (*n* = 7) related to the healthcare system and the way it was organized, and/or to financial barriers.

#### 3.5.2. Studies with Statistical Hypothesis Testing between KBAB and Own Vaccination and Patient Recommendation

When it came to data sources (*n* = 3), those of an official origin led to good own vaccination and patient recommendation practices, whereas those stemming from the mass media or involving disinformation had the contrary effect.

Believing in the efficacy or benefits of the vaccine (*n* = 4) and considering it to be safe (*n* = 3) were the factors most closely related with acceptance of the influenza vaccine. Hence, believing the vaccine to be safe and effective was positively linked to own vaccination and patient recommendation, and doubting its safety or efficacy was associated with lower acceptance of the vaccine.

The following were also associated with favorable behavior in own vaccination and patient recommendation: (1) perceived risk of contracting the disease or experiencing a severe form of it; (2) trust in the health authorities; (3) changing vaccine efficacy from year to year, healthcare organization and patients, all of which acted as external factors; and (4) previous practices or favorable behaviors with respect to other vaccines and past history of vaccination.

More advanced ages and group practices showed a statistically positive correlation with higher vaccination rates among PCP. Having longer clinical experience and a greater burden of care were also associated with favorable results in own vaccination and patient recommendation, whereas occasional practice of alternative medicine decreased vaccination.

### 3.6. HPV Vaccines

#### Studies without Statistical Hypothesis Testing between KBAB and Own Vaccination and Patient Recommendation

All the papers that furnished information on HPV vaccine and KBAB (*n* = 6) were descriptive. Most physicians consulted official data-sources or scientific journals (*n* = 4), and some PCP were confused about the guidelines due to lack of information. Despite the variability of knowledge evaluated by each study in response to the questions posed, all highlighted shortfalls (*n* = 4), particularly when it came to vaccination counselling afforded to older and/or male patients versus young woman patients.

In general, while PCP did not fear the adverse effects of vaccine, 10% to 37% displayed uneasiness about its safety in three studies. In these same papers, most of the participants acknowledged the benefits and efficacy of the vaccine in the prevention of HPV-related diseases.

The studies that described communication with patients about vaccination (*n* = 4) reflected good practices. In other words, most PCP discussed the benefits and risks of the HPV vaccine, informed young women about it, and collected data on sexual practices.

### 3.7. COVID-19 Vaccines

We found three papers (one of them descriptive) that assessed vaccination against SARS-CoV-2, the cause of the current COVID-19 pandemic. In all the articles, participants reported concerns about the safety of vaccines due to their rapid development in a pandemic situation, as well as lack of trust in the health authorities and the information they provide. Finally, low perceived risk among those who consider that the pandemic is not serious and have not received influenza vaccine in previous practices are also related to lower acceptance of COVID-19 vaccines.

## 4. Discussion

Currently one of the main public health challenges is vaccine hesitancy, particularly in the adult population. PCP can play a decisive role in encouraging immunization among their patients because they are not only the main professionals offering vaccine counselling to them, but also their most important source of information. To explain the own vaccination and patient recommendation process, we proposed a modified KAP model (Figure 2).

When it comes to deciding on their own vaccination or on counselling and making recommendations to patients, physicians’ lack of the necessary knowledge is an obstacle that can, in great measure, hinder the process [26,33,45,48,56,59]. To obtain the proper knowledge, it is essential to have access to data sources based on scientific evidence or official resources (EMA, CDC, FDA), which are linked to with higher own vaccination and patient recommendation [24,29,31,33,34,64,65].

According to our model, beliefs are formed based on knowledge, and it therefore follows that if the latter is inappropriate or inadequate, the resulting beliefs are not going to be accurate. Believing that the vaccine is safe [24,28,34] and effective [28,34], and trusting in health institutions [24,28,34] are positively related with own vaccination and patient recommendation, as are being aware of the risk [28,34] and feeling responsible for the vaccination process [24]. Since hesitancy is not stable, strategies can be put in place to reduce it; combating misinformation in networks and the media; informing the process of vaccine development; and increasing credibility in the institutions.

Interestingly, very few studies included in our review displayed an association between socio-demographic factors and own vaccination and patient recommendation, something that is relevant, given that these are not modifiable factors. These results are in line with those of other reviews of physicians’ behavior in matters such as notification of adverse drug reactions [66] or prescription of antibiotics [67]. Only the practice of alternative medicine (homeopathy, acupuncture, etc.) was negatively associated with own vaccination and patient recommendation in a number of studies [24,32,34,54,55,57]. It could be explained because the practice of alternative medicine brings with it a distrustful attitude toward conventional medicine and concerns about the safety of medicines. In addition, disinformation and anti-vaccine campaigns often recommend alternative practices [68].

Our analysis highlights the fact that there are other extrinsic factors, proposed in the modified KAP model, which influence own vaccination and patient recommendation. Hence, there are a number of coexisting determinants that are related with: the patient (baseline level of vaccination hesitancy and resistance to vaccination) [69]. It is therefore important to implement: (1) physician communication training, essentially equipping PCP with the necessary persuasive skills to enable them to convey the importance of vaccination to their patients; and (2) awareness-raising campaigns and educational programs targeted at the entire population, which could serve to counteract the disinformation published by anti-vaccine groups [70,71,72].

Lastly, according to our model, the health outcomes (effectiveness, non-prevention of the disease, adverse reactions) observed in previous vaccination campaigns, generate clinical experiences [22,27] for PCP which may be decisive, on interfering -positively or negatively- in KBAB and, by extension, in ensuing behaviors [24,26,28,29,30,32,33] (Figure 2).

To date, very few specific interventions have been undertaken to reduce vaccine hesitancy on the part of PCP [73,74,75] but none of these has yielded the benefits that were sought [76]. In order for interventions to be effective, they will probably have to be purpose-designed on the basis of gaps identified in the health professionals working in this area [77,78,79]. An improvement in professionals’ communication skills might also enhance the effectiveness of such interventions, as has been shown in other settings [80].

Some of the limitations of our review stem from the quality of the studies included: (1) most of the studies only furnish descriptive information, without performing an analysis of the statistical association between KBAB and own vaccination and patient recommendation (*n* = 25); (2) the fact that the outcome measure (vaccination) was questionnaire- as opposed to record-based, implies that there may be a misclassification risk; (3) use of questionnaires which are not fully validated means that important factors may not have been included in the questionnaires, or that there may be misclassification in the exposures; and (4) several studies did not have a satisfactory response rate or sample size and failed to compare their participants against subjects who did not participate so as to assess the risk of selection bias.

Insofar as the review itself is concerned, the main limitation is the heterogeneity present in the definition of the variables, and the way in which these were measured by the component studies, something that makes it impossible to conduct a meta-analysis, and means that on many occasions, when the studies did not define the variables, this work was done by the authors. Future studies will be needed to study the influence of knowledge, beliefs, attitudes and barriers on adult vaccination, and in which the dependent variable (full adult vaccination yes/no) is based on vaccination records, and in which the questionnaires used are fully validated, as has already been done successfully in other domains, such as determinants of antibiotic prescribing, antibiotic dispensing or adverse reaction reporting [67,81].

Main model of attitudes, knowledge and practices, modified by the conditioning factors of vaccination in our systematic review.

## 5. Conclusions

The own vaccination and patient recommendation decision is a complex process in which multiple factors come into play. The own vaccination and patient recommendation decision is a complex process in which multiple factors come into play. This study is a first step to identify those factors, which are responsible for increasing vaccine hesitancy in society and among health professionals. Now, it is time to take action on each of the identified factors: by facilitating access to accurate information about vaccines on the internet, by involving PCPs in setting recommendations, by strengthening their communication skills and reminder systems, and developing specific multi-component interventions for their training.

## Figures and Tables

**Figure 1 ijerph-19-13872-f001:**
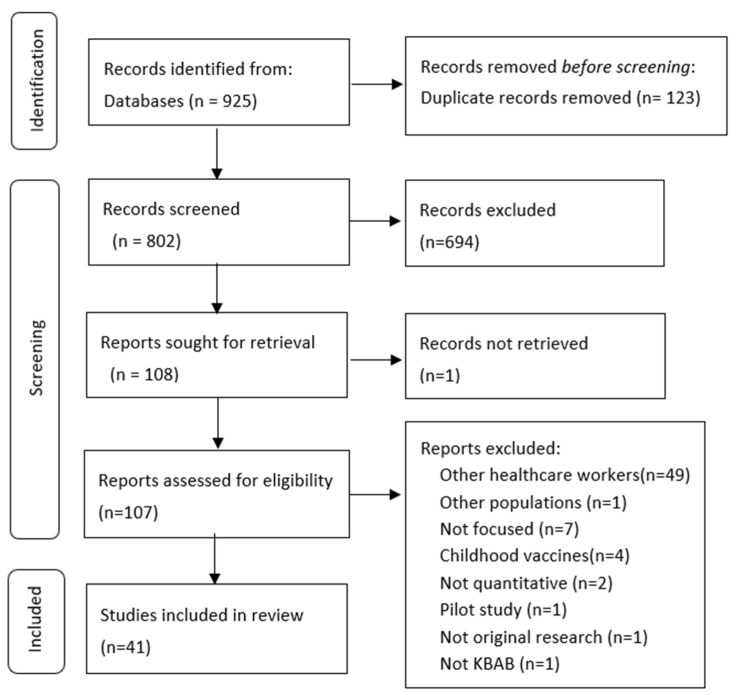
Flow diagram of literature-review search process.

**Figure 2 ijerph-19-13872-f002:**
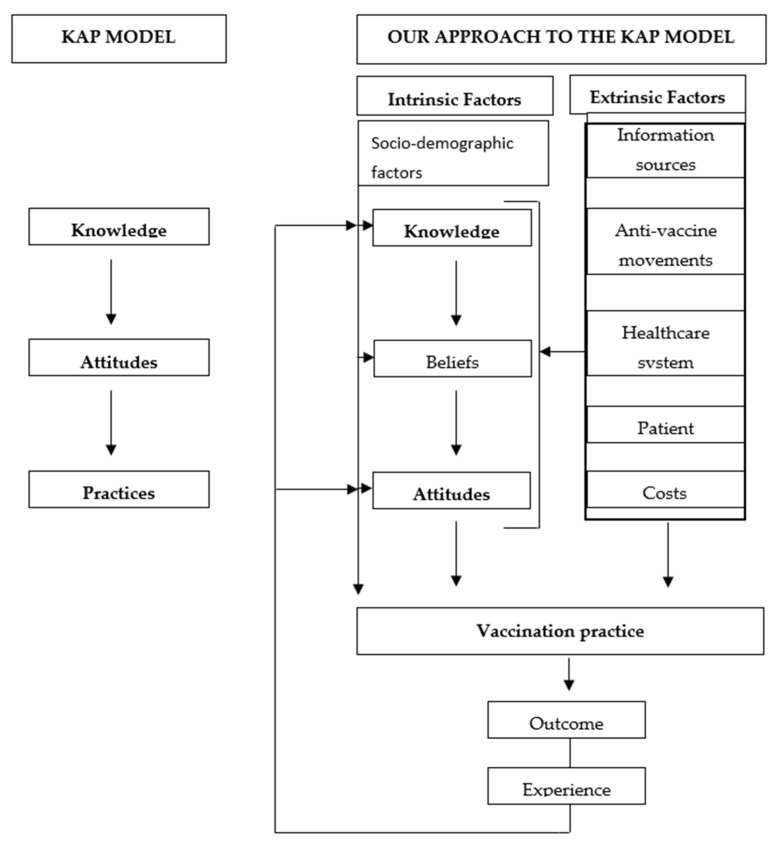
Theoretical framework of the influence of and interconnection among factors that influence vaccination practices (based on A. Teixeira Rodrigues et al.’s model).

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
