# Peer review of "Understanding Primary Care Physician Vaccination Behaviour: A Systematic Review"

_ijerph, 2022, doi:10.3390/ijerph192113872_

Round 1
Reviewer 1 Report
Comment to authors
Abstract
Please state some methodology status instead of “inclusion criteria” here. What parts of the primary articles were extracted or evaluated? More information about the data and how it is collected should be stated.
I would like to say that the authors can give a better suggestion in the conclusion part, something real and valuable. It is similar to the result part.
Keywords
Good.
Introduction
The importance of conducting this study is not well stated in the introduction and the main question should be better explained.
Some health international data related to the topic should be stated.
Materials and Methods
What were the search strategy and the syntax? I cannot find them in the supplementary.
What types of vaccine data were reviewed on the topic of “the vaccination of adults”? all available vaccines? Please list the name of the disease (which their vaccine used in the adult) in the method part.
Studying table number 3 and performing data extraction methods and checking them related to this table is a bit confusing and may be difficult for readers. It is better if it is written with better clarity.
Results
Good, but still could be worked more on the presentation of tables (including Tables 3, sup. 1 and 2) to improve their clarity.
Discussion
Line 334: please check the sentence.
Most of the limitations are declared in the study.
In the end, please suggest a protocol for the future primary study, if possible, to use researchers a similar method for the statistical analysis to evaluate any association between KBAB and own vaccination.
Best regards
Reviewer 2 Report
The paper is interesting to a certain extent. The conclusions need significant widening
Line 149-151 needs an explanation in the manuscript why such a classification was chosen
Some statements need some explanation from the co-authors. For example line 270, The trust placed by PCP in pharmaceutical companies or in reported data was found to be scant because to our mind... or it can be explained by.....
Such explanations give more understanding to readers. In a similar way to widen a sentence (idea ) in lines 345-347
Round 2
Reviewer 1 Report
- Syntax paragraph: please check and revise the final part of the syntax; there are a few extra repeated ‘OR’ here: …AND (“cross-sectional OR” OR “survey” OR “cohort OR” OR “case-control” OR “OR” OR “adjusted OR”).
- According to this response (Response 7: Yes, we have addressed all vaccines used in adults…..), please add a brief explanation to the text of the method. I am sure that this question will be repeated for other readers without further explanation.
